# Preliminary Study on the GWP Benchmark of Office Buildings in Poland Using the LCA Approach

**Joanna Rucińska** [1,*] **, Anna Komerska** [1] **and Jerzy Kwiatkowski** [2]

[1]  Division of Air Conditioning and Heating, Faculty of Building Services, Hydro and Environmental Engineering, Warsaw University of Technology, Nowowiejska 20, 00-653 Warsaw, Poland; anna.komerska@pw.edu.pl

[2]  District Heating and Natural Gas Systems Division, Faculty of Building Services, Hydro and Environmental Engineering, Warsaw University of Technology, Nowowiejska 20, 00-653 Warsaw, Poland; jerzy.kwiatkowski@pw.edu.pl

*  Correspondence: joanna.rucinska@pw.edu.pl

**Abstract:** The decarbonisation goal stated in the Energy Performance of Buildings Directive (EPBD) regarding the building sector will be achieved only if the whole building life-cycle is considered. To fulfil this requirement, a benchmark based on the life cycle assessment (LCA) must be integrated into the early planning phase of buildings by designers. The estimation of such indicators requires the development of a database of building assessments. In this study, an LCA of 11 office buildings in Poland was used to set average values that can be used as a benchmark. The LCA methodology based on the Building Research Establishment Environmental Assessment Method (BREEAM) certification was used. The analysis did not concentrate on one type of office building. The main objective was to investigate a possible range of total Global Warming Potential (GWP) index values normalized to the usable unit floor area. The importance of the GWP of individual life-cycle phases was also considered. The study shows that the used methodology is adequate for LCA benchmark estimation to set preliminary average values for office buildings in Poland.

**Keywords:** LCA; benchmark; building certification; circular economy; environmental impact

## 1. Introduction

Energy consumption in buildings is responsible for 35% of greenhouse gas (GHG) emissions worldwide. If the whole building industry sector is included, emissions increase up to 65% [1]. The world's population is still growing, and new buildings are needed. The issue of reducing energy consumption and GHG emissions from the building sector is one of the most urgent challenges facing the sector today.

From 2002, actions focused on decreasing energy consumption and, as a result, GHG emissions from buildings were taken. The Energy Performance of Buildings Directive (EPBD) [2] introduced the system of energy performance certificate system which is used to assess energy use in buildings. The certification scheme was introduced to increase the awareness of building owners and users of their energy consumption and to ignite a process to increase building energy performance. In 2010, the recast of the EPBD [3] introduced the concept of a nearly zero energy building (nZEB) as "*a building that has a very high energy performance ( . . . ). The nearly zero or very low amount of energy required should be covered to a very significant extent by energy from renewable sources, including energy from renewable sources produced on-site or nearby*". According to the directive that will be put into place on 31 December 2020, all new buildings must be nZEB. EU countries have implemented provisions of the directive in their national building regulations. The means of expressing requirements of buildings' energy

performance has been left to the decision of member countries. In the revised EPDB from 2018 [4], a goal of reducing greenhouse gas emissions by at least 40% of the 1990 value by 2030 has been set. Also, a more ambitious decarbonisation challenge of building stock by 2050 has been established.

To meet such requirements, more complex solutions than just a reduction of energy at the operational stage must be applied. One of the methods is to introduce the circular economy concept (CE). This idea is aimed at the continual use of resources, and by this, eliminating waste generation. The concept of circularity is based on the creation of a closed-loop system, reduction of resource use and waste creation, and finally, decreases in pollution and carbon emissions [5]. This has often been described as the 3Rs: Reduce, reuse, and recycle [6]. However, research done by Nuñez-Cacho et al. [7] showed that more dimensions are needed to describe the CE scale for the building industry. Besides the 3Rs, three other dimensions related to resource management have been determined: energy, water, and materials management efficiency. Two environmentally characterized dimensions have been described—emissions and waste generation—and, finally, one indicator representing the transition to the circular economy has been added. The building sector is one of the most significant emitters of pollutants, generators of waste, and users of resources [8]; therefore, the implementation of the CE concept in the building industry is so important. The implementation should not be limited to material manufacturing but should also include construction companies, where it will contribute to the creation of jobs and economic growth [9].

The integration of the circular economy concept with decarbonisation requirements from EPDB can be done by using the Life Cycle Assessment methodology (LCA) to determine a building's total GHG emissions throughout its lifetime. LCA is used to assess the environmental impact at all stages of the life cycle, from raw material extraction, manufacturing, distribution, and use, to recycling/reuse and final disposal at the end [10]. By using LCA, both the embodied energy and operating energy are taken into consideration. The embodied energy in buildings can be divided into the initial (IEE) and recurring embodied energy (REE) [11]. The IEE represents the non-renewable energy used for raw material extraction, processing, manufacturing, transportation and construction. It is composed of two components: direct energy, which is used for product transportation on-site and then building construction and indirect energy, used to acquire, process, and manufacture the building materials, and transportation needed for these activities. The REE can be described as the non-renewable energy used to maintain, repair, restore, refurbish, or replace materials, components, or systems over the life cycle of the building. In Figure 1, a qualitative representation of the energy consumption of residential buildings in China over the building life span (50 years) is presented [12].

At the beginning of a building's life, the embodied energy related to construction materials dominates; however, with subsequent years of use of the building, its operational energy exceeds its EE value. The boundaries of the life cycle assessment are determined by its scope. Lausselet et al. used the life cycle assessment to estimate GHG emissions for zero energy neighbourhoods [13]. In the analysis, elements like building, mobility, open spaces, networks, and on-site energy generation were taken into consideration. It was shown that buildings represent the majority (52%) of total GHG emissions, closely followed by mobility (40%). In most of the cases, the commuting energy is neglected, and only environmental impacts related to building materials, the construction and operation stage, and building disposal are included in the LCA according to EN15978 [14]. An evaluation framework for environmental impact assessment tools used in the early stages of the building design was presented by Mexx et al. [15]. It was shown that LCA tools can be very useful for concept design planning where the first decisions on architectural design are made. However, some challenges must be overcome to make them more user-friendly, like simplification of the calculation methodology and integration of the architects' work methods.

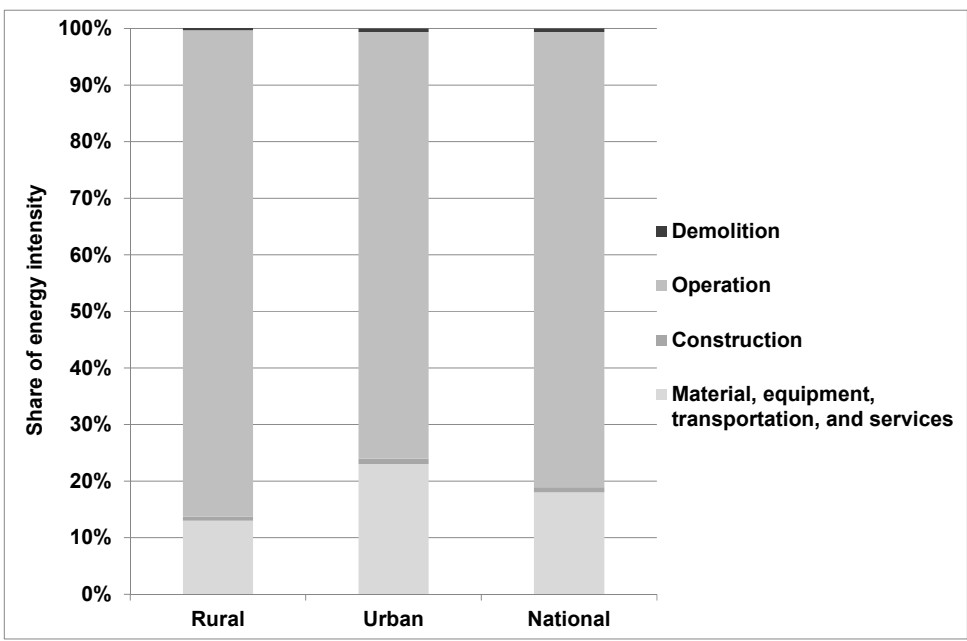

**Figure 1.** Share of energy intensity over different life stages of residential buildings in China.

Currently, there is no obligation to use LCA methodology, but it is sometimes used to assess the environmental impact in the early stages of building design. In many studies [16–20], the LCA methodology has been used to show comprehensive optimization of a building's design solution and thus to assess its carbon footprint. Some studies have focused on the influence of a single parameter on total GHG emissions, like [21,22], where the effect of the window to wall ratio (WWR) of office buildings on the equivalent $CO_2$ emissions was analysed. It was shown that the total carbon footprint is greater for buildings with a higher WWR, which is mostly related to increased energy demand. In other research, the impact of different design solutions of roof-mounted PV systems on the GHG emissions of zero-emission buildings in Norway was presented [23].

Bearing in mind the goal of decarbonisation of the construction sector set out in the EPBD, it seems logical to establish requirements for greenhouse gas emissions for newly designed buildings. The EPBD forced EU countries to implement the requirements of nZEB into national building laws. However, the definition of nZEB refers only to energy needs in the use phase of a building's lifetime. Based on many case studies, it has been acknowledged that, in conventional buildings, most of the energy and GHG emissions production is related to the operational stage [19,24,25]. In [24], a summary of 73 case studies made for 13 countries was presented. The study showed that operational energy contributes to 80–90% of the total energy produced during the life cycle and, therefore, has a major influence on a building's environmental impact. It was also shown that 10–20% of embodied energy is related to the product stage, while the rest of the building phases have a negligible or little share. For buildings with better energy performance (low energy and nZEB), due to significant reductions in energy needs, the embodied energy and embodied carbon represent more significant shares in the whole life cycle [20,25]. The GHG emissions related to the operational stage are directly dependent on the specific energy mix of the country. Therefore, if renewable or low carbon fuels are considered, the total contribution of the construction materials to GHG emissions during a building's lifetime could rise to 80% [18]. As the influence of material emissions is becoming more significant [18,26], further improvements, in terms of carbon footprint reduction, should also be considered in the product stage.

The integration of all stages of a building's lifetime in $CO_2$eq emission requirements for new buildings is reasonable. In this way, the building's environmental impact will be considered at every stage. However, to integrate the carbon footprint into building assessment, an analysis of the current state is needed. In this paper, a preliminary GHG emission benchmark for office buildings

in Poland is presented. Similar studies have been presented by Schegl et al. [27] for German office buildings, Lasvaux et al. [28] for low-energy single-family houses in France, and Passer et al. [29] for five low-energy houses in Austria; however, due to the different building types or, what is even more important, the different energy mix of countries, the results cannot simply be adopted in other countries. The analysis is provided based on data from 11 office buildings that were assessed according to the Building Research Establishment Environmental Assessment Method (BREEAM) certification. Energy consumption was calculated using dynamic simulations, and the LCA results comply with BREEAM requirements. The estimated carbon footprint benchmark defines the state-of-the-art situation and describes the average values.

## 2. Materials and Methods

### 2.1. Energy Demand in the Operational Phase

Depending on the building type and scope of the analysis, an appropriate simulation tool should be selected. The energy need for this study was calculated using the dynamic modelling software Design Builder 4.7.0.027 [30]. To calculate the energy demand for heating, cooling, domestic hot water, lighting, and auxiliary appliances for building, it is necessary to define the following input data: building geometry and zoning, construction of internal and external partitions (with the definition of characteristic parameters of materials), shading devices, weather data, internal gains (appliances and processes), occupancy, airtightness level, ventilation rate, and heating and cooling setpoints. Besides the information on building elements and heating, ventilation and air conditioning (HVAC) systems, data on users' clothing (clothing insulation), activity level, and metabolic rate must also be specified. Based on the results of the energy need calculation and the efficiency of the systems, the energy use was estimated. The efficiency of the systems was determined by the losses of energy generation, transmission, regulation and its use as well as storage. Finally, based on the energy use demand and considering the non-renewable primary energy factor for energy sources, the primary energy demand was calculated. The analysis was carried out under Polish building law; therefore, the calculations include only the energy demand of the systems that are considered in the energy performance certificate of buildings. For the above reasons, energy use related to office equipment and elevators was excluded from the analysis. Polish regulations, which have changed in recent years, also specify a maximum thermal transmittance of external partitions and maximal primary energy demand indicator (Table 1).

**Table 1.** Changes in Polish technical requirements.

| Parameter | From 1st January 2009 | From 1st January 2014 | From 1st January 2017 |
|---|---|---|---|
| Thermal transmittance *U*-value [W/(m²K)] | | | |
| External walls | 0.30 | 0.25 | 0.23 |
| Roof | 0.25 | 0.20 | 0.18 |
| Ceiling above the basement | 0.45 | 0.25 | 0.25 |
| Windows | 1.8 | 1.3 | 1.1 |
| Solar energy transmittance *g*-value [-] | 0.5 | 0.35 | 0.35 |
| Primary energy demand indicator [kWh/(m²year)] | depending on the building shape ratio A/V [1] | 190 | 185 |

[1] Building shape factor A/V, where A is the area sum of all external surfaces of a building in contact with ambient air and V is the heated or cooled building volume.

### 2.2. Emission Level throughout a Building's Life Cycle

The evaluation of the environmental impact was performed according to the set of ISO standards concerning the Life Cycle Assessment approach (ISO 14040–14044) [31,32]. According to these standards, the LCA is composed of four steps: goal and scope definition; inventory analysis; life cycle impact assessment (LCIA); and interpretation. This integrated methodology enables the environmental

impact of a product, buildings, or process to be quantified throughout its entire life cycle. It is therefore one of the most recognized standards of LCA study [33]. Life Cycle Assessment was performed using the calculation tool OneClick LCA, which is compliant with European standards.

### 2.2.1. Goal and Scope Definition

#### Aim of the Analysis

The main goal of the study was to perform an LCA study for a preliminary proposal of a possible emission benchmark for newly built office buildings located in Poland. The analysis did not concentrate on a particular type of office building or a given construction method, materials, HVAC system, or building elements considered in the life cycle inventory. The main objective was to investigate the differences and a possible range of total Global Warming Potential (GWP) index values normalized to the usable unit floor area. All analysed buildings were constructed between 2009 and 2019 according to Polish standards and energy requirements. The lease office area was assumed to be finished to the shell and core standard, i.e., without partitions, which is a common business practice in Poland. The life cycle assessment was performed for a reference building's lifetime, assumed to be 60 years. The goal and scope definition defined for the LCA are summarised in Table 2.

**Table 2.** Summary of the goal and scope definition.

| Goal | Life Cycle Analysis to Define a Benchmark Based on the Total GWP | |
|---|---|---|
| | Building type | Office |
| | Structure | Concrete structure |
| | Location | Poland |
| Functional equivalent | Finishing standard | Shell and core |
| | Assumed reference building lifetime TLT (years) | 60 |
| | Construction period | 2009–2019 |
| | Functional unit | 1 m$^2$ of usable floor area |
| | Assessed impact categories | Global Warming Potential (GWP) |
| System boundary | Cradle to gate Building construction materials excluding technical equipment and furnishingsEnergy in operation stage excluding office equipment, elevators, and commuting energy | |
| Calculation Software | One Click LCA© and 360optimi, Bionova [34] | |

#### System Boundaries

The environmental impact is associated with different stages of a building (i.e., from raw material extraction through to materials processing, manufacture, distribution, use, repair and maintenance, and disposal or recycling). The LCA for the analysed office buildings was performed for building life stages, according to the European standard EN15978 [14]. Although some studies [18,19] have concluded that the construction and end-of-life stages have minor influences on the total GWP index value, especially for typical buildings not defined as nearly Zero-Energy Buildings (nZEB) or Green Buildings, the analysis was performed for all building stages to give a more comprehensive analysis. The environmental-impact-related phases B1–B3 were not included in the study due to a lack of specific data. According to EN15978, the total Global Warming Potential of the system was calculated for stages A, B, and C. Stage D, not included in the totals, shows, however, potential benefits and loads beyond the system boundary, resulting from material or energy recovery from recycling or material reuse. A graphic interpretation of the system boundary is shown in Figure 2.

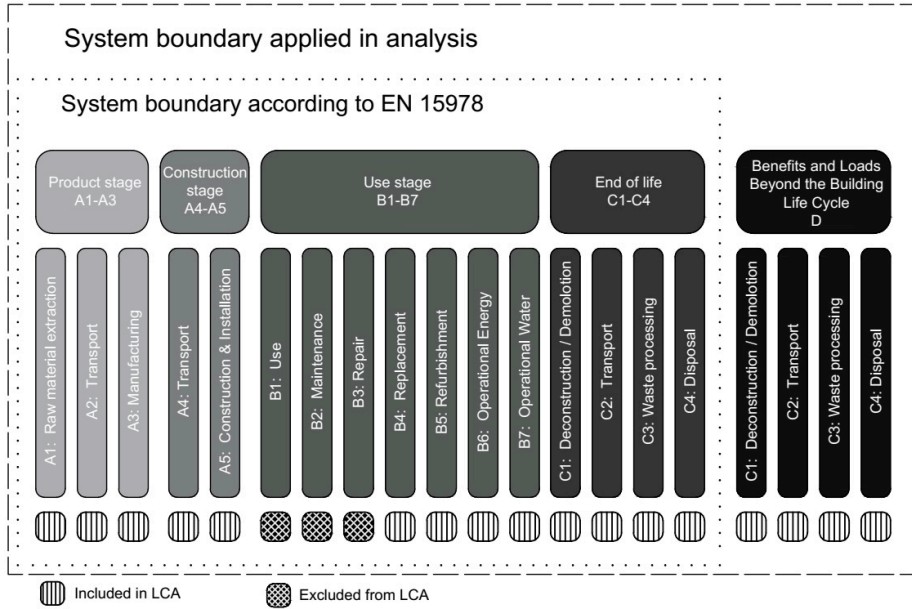

**Figure 2.** System boundary definition.

### 2.2.2. Life-Cycle Inventory

According to ISO 14041, the life cycle inventory involves the collection of the required data and the calculation of related inputs and outputs, such as material types and quantities, energy use, transport distances, and emissions. This part of the LCA study is considered to be the most crucial, as the final results and accuracy directly depend on the quality of the collected data. For each of the analysed buildings, a separate life cycle inventory (LCI) was created, and all inputs and outputs were categorised according to the building life cycle stage, as follows: production phase, construction phase, use phase, end of life and stage D—Potential benefits and loads beyond the system boundary. The assumptions applied to the LCA study are summarised in Table 3.

**Table 3.** Summary of Life Cycle Assessment (LCA) study assumptions.

| Area of Analysis | Data Sources and Assumptions |
| --- | --- |
| Material quantities (A1–A3) | Data inventory based on technical documentation provided in the Executive Design for Architecture and Structure. Material database: EPD, Ecoinvent, Okobau.dat, Bionova. Local material manufacturer data if possible. Otherwise, data of regional European average material were selected |
| Building material transport distances (A4) | Transport distances were estimated based on the average European transport distance specific for each material type |
| Construction and installation process (A5) | Average emissions for the construction process were based on the project size using a scenario provided in the calculation tool |
| Material service lifetime (B4–B5) | The material service life was based on data related to the specific material or specific project values when available. Otherwise, default values from the One Click LCA database were applied |
| Building use phase energy consumption (B6) | Energy consumption was based on energy simulations. For grid electricity and district heating, emissions were calculated according to the Polish energy mix |
| Building use phase water consumption (B7) | Water consumption was estimated on the basis of the average water consumption for utility purposes and for watering green areas according to Polish legislation |
| End-of-Life Stage (C1–C4) | Based on a scenario provided in the calculation tool |
| Benefits and loads beyond the building life cycle (D) | Based on a scenario provided in the calculation tool |

Material Production Phase A1–A3

All analysed buildings subjected to LCA were real office buildings constructed in Poland between 2009 and 2019. Material types and quantities were derived from technical documentation provided in the buildings' executive designs for architecture and construction. This approach ensured a high level of accuracy in the collected data inventory and provided very detailed information about the analysed buildings, starting from concrete reinforcement densities up to the finishing types. Materials and products were categorised according to the building element type compliant with the BREEAM standard: external walls (envelope, structure and finishes), external windows and rooflights, foundations, internal floor finishes, structural frame, upper floors, basements/retaining walls, ground/lowest floor, internal ceiling finishes, internal walls and partitions, roof (including coverings), stairs and ramps, balustrades and handrails, internal doors, internal wall finishes, internal windows and hard landscaping. HVAC equipment as well as technical systems were excluded from the assessment.

The specific type of material or product was selected from a material database provided in the calculation tool. The local manufacturer was selected if possible. Otherwise, either regional-specific or average European data were chosen, and, consequently, emissions were compensated according to the local Polish energy mix. The LCA data of all selected materials or products were assessed using publicly available and peer-reviewed methodology/PCR, compliant with EN 15804 [35].

Construction Phase A4–A5

The construction phase refers to all energy consumed on-site during building construction. The category includes fuel for the machinery and building equipment as well as energy use for heating and cooling of office rooms. Due to a lack of precise data, the same general scenario for construction site operations, available in the calculation tool, was selected for all analysed buildings.

Materials and products transported from the manufacturer to the construction site were calculated in stage A4. Transport distances were estimated based on the average European transport distance specific for each material type. The three following transport scenarios were adopted:

- A concrete mixer with a capacity of 8 $m^3$ and 100% filling for concrete transportation;
- A truck with a 40-ton capacity and a 100% fill rate for transportation of large-scale and large-volume materials, e.g., reinforcing steel, concrete blocks, insulation, glass facade modules, and insulation;
- A dumper truck with a capacity of 19 tons with a 100% fill rate for transportation of loose materials, e.g., gravel, sand, and soil substrate.

Material Service Lifetime (B4–B5)

During the building life cycle, some materials will need to be replaced after a certain period of time. The structure of all buildings made of reinforced concrete was assumed to be permanent and, therefore, a building's lifetime was defined according to the reference operation time. For remaining materials, the replacement periods were based on specific data provided in the Environmental Product Declaration (EPD), if available. Otherwise, if generic data were selected, the material or product service lifetime was defined based on its type according to the following scenario: windows and glass façade—30 years, insulation—30 years, internal and external doors and garage gates—30 years, foils and membranes—20 years, and exterior plaster coating and finishing materials—30 years.

Operational Energy and Water Use (B6–B7)

Energy consumption was obtained through energy simulations performed with comprehensive thermal analysis software—DesignBuilder. More detailed information about the calculation procedure used and assumptions made are described in Section 3.1. The GWP index for energy sources was defined according to a calculation tool [34], as follows: grid electricity for Polish energy mix GWP =

0.936 kgCO$_2$eq/kWh, heating network (average) GWP = 0.474 kgCO$_2$eq/kWh, and natural gas GWP = 0.244 kgCO$_2$eq/kWh.

Water consumption was estimated based on the average water consumption for utility purposes and watering green areas, according to Polish legislation [36].

### 2.3. Benchmark Procedure

The noun "benchmark" can have different meanings, for example, a level of quality that can be used as a standard when comparing other things, or criteria that a product is expected to meet, or a point of reference by which something can be measured. A benchmark can be described as a limit value, reference value, best practice value, or target value [37]. The limit value is the lowest acceptable value that must be reached. Otherwise, it has to be decided whether the assessment is still valid. The present state-of-the-art standard is described by the reference value. It can be considered a median or average value. The best practice value reflects the best value that can be achieved experimentally. The target value represents the highest theoretical value that can be reached. The estimated GWP benchmarks in this article define the state-of-the-art standard and describe the average values. The objective of this study was to preliminarily define building classes that can be used by architects to assess new developments. The choice of reference value methodology is consistent with the rating method used to assess the energy performance of buildings described in the European standard EN ISO 52003-1 [38].

The benchmark procedure was performed in relation to the Global Warming Potential factor, which was specified for all buildings. For the collected dataset, the standard deviation of the GWP index was calculated to investigate the possible correlation or diversity between analysed buildings. Statistical analyses were performed to investigate the distribution of the dataset and to detect possible extreme results. If any of the data expressed in the normalised GWP index significantly deviated from the dataset, they were regarded as abnormal and rejected from the benchmark definition [39].

### 2.4. Assumptions and Simplifications

The authors of this study intended to provide highly accurate data through the development of a detailed data inventory and comprehensive energy simulation calculations. Nevertheless, a few limitations and uncertainties resulting from the necessary assumptions and still-limited database were identified, as follows:

- The accuracy of the material specifications: This uncertainty should not be significant as the environmental impact of all selected materials and products was calculated according to the same methodology. Moreover, if a local manufacturer was not available in the database, the GWP index was compensated with the local Polish energy mix;
- General scenarios were applied for the construction phase, end of life phase, and stage D. Transport distances were based on the typical average European transport distances, assuming the vehicles were filled with 100% cargo materials. However, as the construction and end-of-life stage were proven to have minor influences on the final environmental impact in the building total life cycle, the accuracy of the assumed simplifications was defined as satisfactory;
- Results were normalized to the usable floor area. This assumption might have resulted in a minor under or overestimation of the normalised GWP index as buildings are characterised by different underground level areas.

## 3. Description of the Building Database

### 3.1. General Information

For the purposes of the analysis, a database consisting of 11 office buildings located in Poland was created. The buildings selected for the analysis varied in terms of characteristic parameters such as their usable area, number of floors, shape, type of windows, glazing ratio, HVAC system, and energy sources. Although the created database is limited to 11 buildings, it reflects a variety of typologies

and technologies present in real buildings. The characteristic parameters are summarised in Table 4. The buildings were built in the last 10 years. In this time, the requirements for the thermal protection of buildings have changed several times: In 2009, 2014, and 2017. The Polish climate is specified according to five climate zones for winter (from I—least cold to V—the coldest) and two for summer (I—warm and II—hot).

**Table 4.** General buildings information.

| Parameter | B1 | B2 | B3 | B4 | B5 | B6 | B7 | B8 | B9 | B10 | B11 |
|---|---|---|---|---|---|---|---|---|---|---|---|
| Total area (incl. garage) [m$^2$] | 25,639 | 39,349 | 10,251 | 23,913 | 11,337 | 14,969 | 67,672 | 37,598 | 20,011 | 28,336 | 25,667 |
| Usable area [m$^2$] | 17,962 | 24,324 | 5832 | 17,515 | 8541 | 12,093 | 51,270 | 31,122 | 13,285 | 15,750 | 18,542 |
| Polish building requirements | 2017 | 2017 | 2017 | 2017 | 2017 | 2009 | 2014 | 2017 | 2017 | 2014 | 2014 |
| Number of storeys (incl. garage) | 6 | 8 | 5 | 7 | 8 | 8 | 39 | 18 | 6 | 9 | 10 |
| Number of aboveground storeys | 5 | 6 | 4 | 6 | 7 | 7 | 36 | 15 | 5 | 7 | 8 |
| A/V | 0.21 | 0.21 | 0.28 | 0.11 | 0.16 | 0.22 | 0.12 | 0.12 | 0.21 | 0.22 | 0.18 |
| WWR | 39.8% | 44.6% | 78.7% | 48.3% | 48.0% | 49.3% | 73.7% | 72.5% | 58.4% | 43.8% | 64.8% |
| Climate zone winter | III | III | III | III | III | III | I | I | III | III | III |
| Climate zone summer | II | II | II | II | II | II | II | II | II | II | II |

Common features and parameters can be distinguished in all of the analysed buildings, as follows: the ratio of the lease area to the total usable area of the building, internal heat gains from people and equipment, the airflow rate per person and how it is adjusted in individual zones, schedules of occupancy, lighting and equipment usage, and temperature and humidity settings in particular zones. All buildings are characterised by a low building shape factor and a window-to-wall ratio varying from 39.8% to 78.7%. Although they are located in different climate zones, the weather data do not differ a lot. The energy use demand indicator, with division into energy carriers, and the primary energy demand indicator, calculated according to the above-described assumptions, are presented in Table 5.

**Table 5.** Energy use and primary energy demand.

| Parameter | B1 | B2 | B3 | B4 | B5 | B6 | B7 | B8 | B9 | B10 | B11 |
|---|---|---|---|---|---|---|---|---|---|---|---|
| Energy use demand indicator kWh/(m$^2$year): | | | | | | | | | | | |
| • District heating | 18.5 | 25.2 | 20.9 | 19.3 | 0.0 | 37.3 | 32.4 | 11.2 | 18.8 | 12.9 | 0.0 |
| • Gas | 0.0 | 0.0 | 0.0 | 0.0 | 132.1 | 0.0 | 0.0 | 0.0 | 0.0 | 0.0 | 40.7 |
| • Electricity | 62.5 | 61.7 | 61.9 | 78.2 | 71.1 | 88.4 | 70.2 | 80.3 | 48.3 | 52.1 | 73.3 |
| Primary energy demand indicator kWh/(m$^2$year) | 207 | 190 | 203 | 244 | 217 | 258 | 227 | 246 | 173 | 170 | 265 |

In older buildings, the energy demand for heating has higher values than in newer buildings. In the case of building B10, a portion of the energy used for heating is covered by electricity. In terms of energy use indicators, building B5 significantly differs from the others, mainly due to the use of trigeneration aggregate as the main energy source.

## 3.2. Building Materials

All analysed buildings are characterised by the same monolithic reinforced concrete structure, which is the most common type of structure among newly constructed high-rise buildings in Poland. Materials (concrete, concrete steel reinforcement) used for all structural elements, such as floor slabs,

foundations, columns, load-bearing vertical structures, beams and roofs, were quantified according to the Structural Executive Design. The lease office area was assumed to be finished to the shell and core standard, i.e., without partitions. The inner walls beyond the lease space were designed as aerated concrete blocks and plasterboard wall systems with steel profiles and glass wool insulation. The finishing materials were defined according to Architectural Executive Design. A list of assessed building elements for each analysed building is summarised in Table 6. It includes all main building parts related to structure and finishing and represents the standard in which the building was completed by the construction company. The collected building database varies in terms of the number and types of elements included in the analysis. It therefore shows possible variations in defining building components and resulting outcomes.

**Table 6.** Summary of assessed building elements.

| Building | B1 | B2 | B3 | B4 | B5 | B6 | B7 | B8 | B9 | B10 | B11 |
|---|---|---|---|---|---|---|---|---|---|---|---|
| Total number of assessed building elements | 15 | 15 | 14 | 14 | 13 | 10 | 9 | 11 | 11 | 11 | 10 |
| External walls (envelope, structure, finishes) | Y | Y | Y | Y | Y | Y | Y | Y | Y | Y | Y |
| External windows and rooflights | Y | Y | Y | Y | Y | Y | Y | Y | Y | Y | Y |
| Foundations | Y | Y | Y | Y | Y | N | Y | Y | Y | Y | Y |
| Internal floor finishes | Y | Y | Y | Y | Y | Y | Y | Y | Y | Y | Y |
| Structural frame (vertical) | Y | Y | Y | Y | Y | Y | Y | Y | Y | Y | Y |
| Upper floors (including horizontal structure) | Y | Y | Y | Y | Y | Y | Y | Y | Y | Y | Y |
| Basements/retaining walls | Y | Y | N | Y | Y | N | N | Y | Y | Y | Y |
| External solar shading devices | N | N | N | N | N | Y | N | N | N | N | N |
| Ground/lowest floor | Y | Y | Y | Y | Y | Y | Y | Y | Y | Y | Y |
| Internal ceiling finishes | Y | Y | Y | Y | N | N | N | N | N | N | N |
| Internal walls and partitions | Y | Y | Y | Y | Y | Y | Y | Y | Y | Y | Y |
| Roof (including coverings) | Y | Y | Y | Y | Y | Y | Y | Y | Y | Y | Y |
| Stairs and ramps | Y | Y | Y | Y | Y | N | N | N | N | N | N |
| Balustrades and handrails | N | N | N | N | N | N | N | N | N | N | N |
| Internal doors | Y | Y | Y | Y | Y | Y | N | Y | N | N | N |
| Internal wall finishes | N | Y | Y | N | N | N | N | N | N | N | N |
| Internal windows | Y | N | Y | N | N | N | N | N | Y | N | N |
| Hard landscaping, roads, paths and paving | Y | Y | N | Y | Y | N | N | N | N | Y | N |
| Hard landscaping, fencing, railings, walls | N | N | N | N | N | N | N | N | N | N | N |

Y (dark grey) means included in the LCI and N (light grey) means not included in the LCI.

### 3.3. Description of HVAC Systems

All of the analysed buildings include commonly used HVAC systems designed to maintain the required internal temperature and air humidity. All buildings are equipped with automation and control systems. A mechanical supply-exhaust ventilation system with highly efficient heat recovery is used. All buildings were divided into zones based on their function and orientation. Such solutions allowed the energy demand to be adjusted depending on the heat losses and heat gains. The HVAC systems were designed in accordance with building law, and consequently, all elements are characterized by a high-efficiency standard. The heating, cooling and electricity sources used in the analysed buildings are listed in Tables 7 and 8.

**Table 7.** Heating sources used in buildings.

| Heating Source | District Heating sub-Station | VRF | Gas Boiler | CHP |
|---|---|---|---|---|
| **Building Indicator** | B1, B2, B3, B4, B6, B7, B8, B9, B10 | B1, B2, B3, B4, B8, B10, B11 | B5, B11 | B5 |

**Table 8.** Cooling sources in buildings.

| Cooling Source | Chillers | VRF | Split and Multi-Split Air Conditioners | Adsorption Chiller |
|---|---|---|---|---|
| **Building Indicator** | B1, B2, B3, B4, B5, B6, B7, B8, B9, B10, B11 | B1, B2, B3, B4, B8, B10, B11 | B5, B11 | B5 |

In most of the buildings, separate systems involving a district heating sub-station and chillers are used as heating and cooling sources, respectively. In some of the buildings, a Variable Refrigeration Flow (VRF) system designed for both heating and cooling is used. In one building, an integrated system involving a combined heat and power unit (CHP) for heating purposes and an adsorption chiller for cooling needs is used. The peak loads are covered by gas boilers or split and multi-split air conditioners. Renewable energy sources are not used in any of the analysed buildings.

## 4. Results and Analysis

### 4.1. Life Cycle Impact Assessment (LCIA)

An evaluation of the building's carbon footprint was performed for the total building life cycle using the methodology described in Section 3.2. In order to obtain reference values, the GWP index was normalized to the usable floor area. Figure 3 presents the distribution of the $CO_2$eq emissions for each analysed case over the total building life cycle. As expected, the GWP index related to energy use represents the most significant share—Between 79% and 93% with an average value of 87% ± 4.2%. The embodied emissions in the product stage were identified as the second-largest contributor and are responsible for 6–11% of the building's carbon footprint. The obtained distribution of the GWP index over the life stages is consistent with findings published by other authors [19,24,25,29], and therefore, it was assumed to be correct at a highly satisfactory level, despite the few simplifications applied in the methodology.

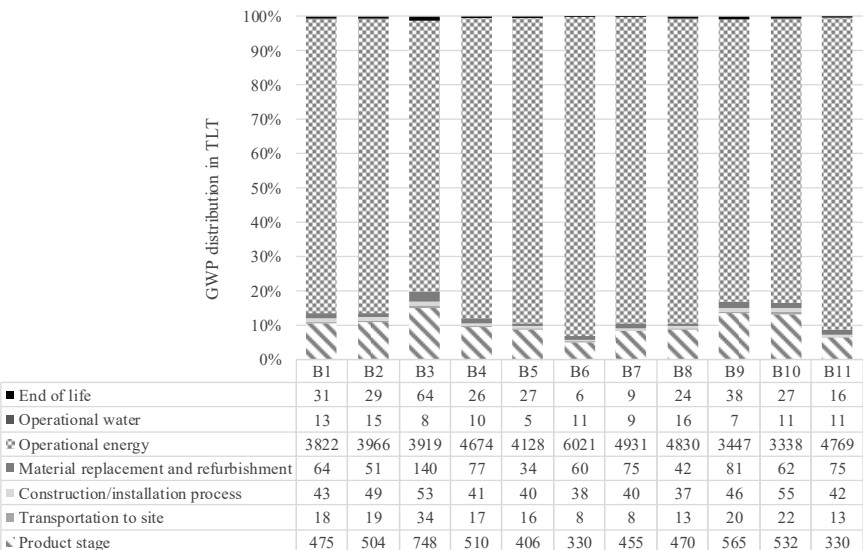

**Figure 3.** Distribution of the Global Warming Potential (GWP) index over a building's life cycle. The values in the table are shown in kgCO$_2$eq/m$^2$ and were calculated for TLT = 60 years.

The level of emissions in the product stage depends on the material quantities and types used. The highest environmental impact is associated with building structures composed of concrete and reinforcement steel, followed by a glazing facade. In the analysed buildings, most of the construction elements were included in the LCI. As the performed life cycle assessment study included most of the contributing materials, it shows the possible range of carbon footprint values over the product stage.

A considerable correlation (presented in Figure 4) was observed between the total mass of building materials and the GWP index, with a regression factor of $R^2 = 0.625$ calculated for all 11 buildings. This shows that, to a certain extent, material quantities play an important role in the product stage of the GWP index. However, other building parameters, such as the building shape, types of elements included in the analysis, and material types also contribute to the emission level and should not be neglected in the analysis.

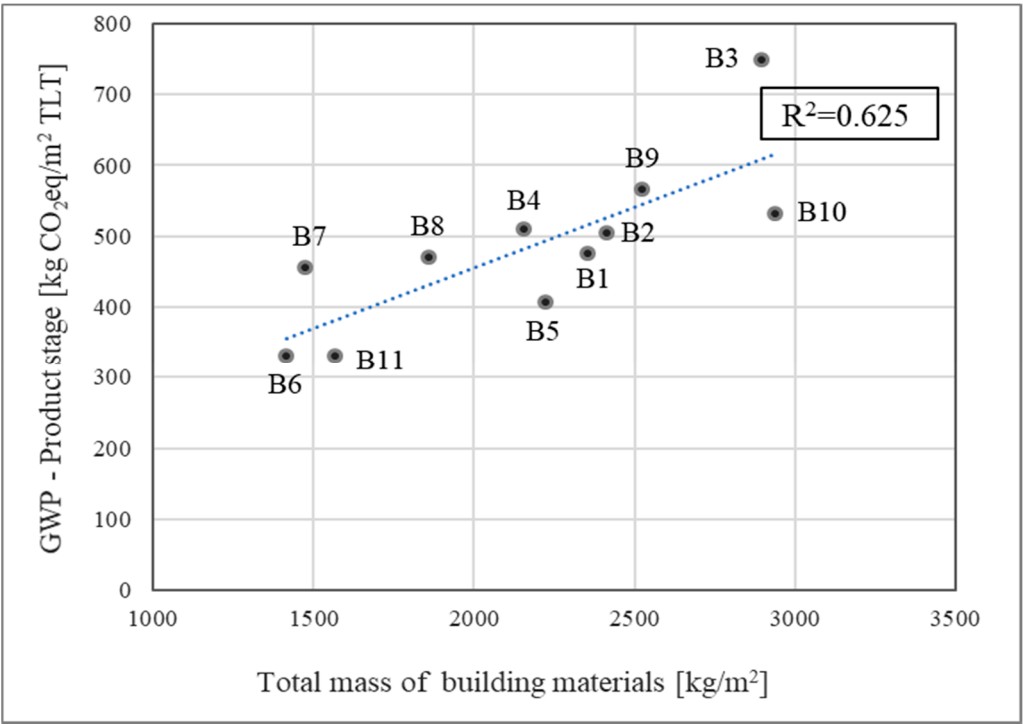

**Figure 4.** Correlation between the building mass and GWP index for the production stage.

Finally, one of the purposes of the circular economy concept is to reduce waste creation through material recycle and reuse. The potential benefits beyond the assessed system boundary are presented in Figure 5. The environmental benefits expressed as the GWP index, result from the reuse and recycling of construction materials or fuels. The transformation process includes waste recycling from concrete, steel and aluminium profiles or the incineration of wood and plastic. Despite the high quantities of those materials embodied in the building structure, especially concrete and reinforcement steel, the level of reduction of normalised $CO_2$eq emissions represents very low values, varying over the analysed buildings between 40 and 110 $kgCO_2eq/m^2TLT$, with an average value of 58 $kgCO_2eq/m^2TLT$. Accordingly, it represents a share in the total lifetime of between 0.6% and 2.2%, meaning that only 9–13% of the embodied materials can be reused or recycled.

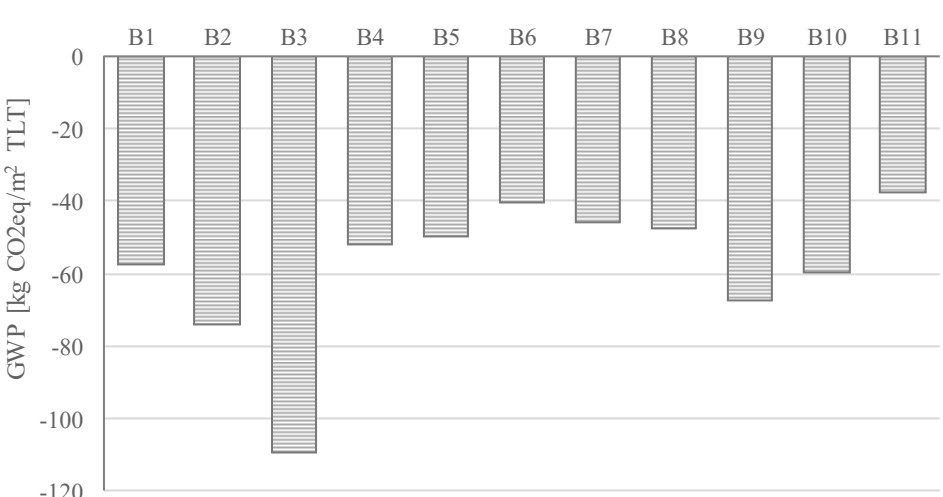

**Figure 5.** Potential benefits beyond the assessed system boundary.

The environmental impact of the assessed buildings, expressed as the GWP index, has quite high values compared with those measured for other countries, e.g., Germany [27]. These differences result from the use of different energy sources in the energy mixes of different countries. Grid electricity as well as district heating is produced in Poland mainly through the combustion of hard and brown coal [40].

*4.2. Defining the Benchmark*

Based on the results of the life cycle impact assessment study, the collected database was analysed to determine a possible benchmark definition for office buildings located across Poland. Although the created building database is not statistically representative of all buildings, it shows the current state and level of Global Warming Potential of office buildings designed and constructed according to current legislation, energy requirements, and technology. Figure 6 presents the mean value and standard deviation derived for all 11 buildings. The mean value for the total GWP index was found to be almost 5000 $kgCO_2eq/m^2TLT$ with a standard deviation of 701 $kgCO_2eq/m^2TLT$. Most of the buildings were found to have values within the defined range, with only 1 building clearly deviating from those values and representing a noticeably higher level of Global Warming Potential. According to the methodology described in Section 2.3, the buildings with the highest and lowest GWP index values were excluded from analysis. This approach limited the number of analysed buildings; however, it excluded extreme data that could have wrongly influenced the final outcome.

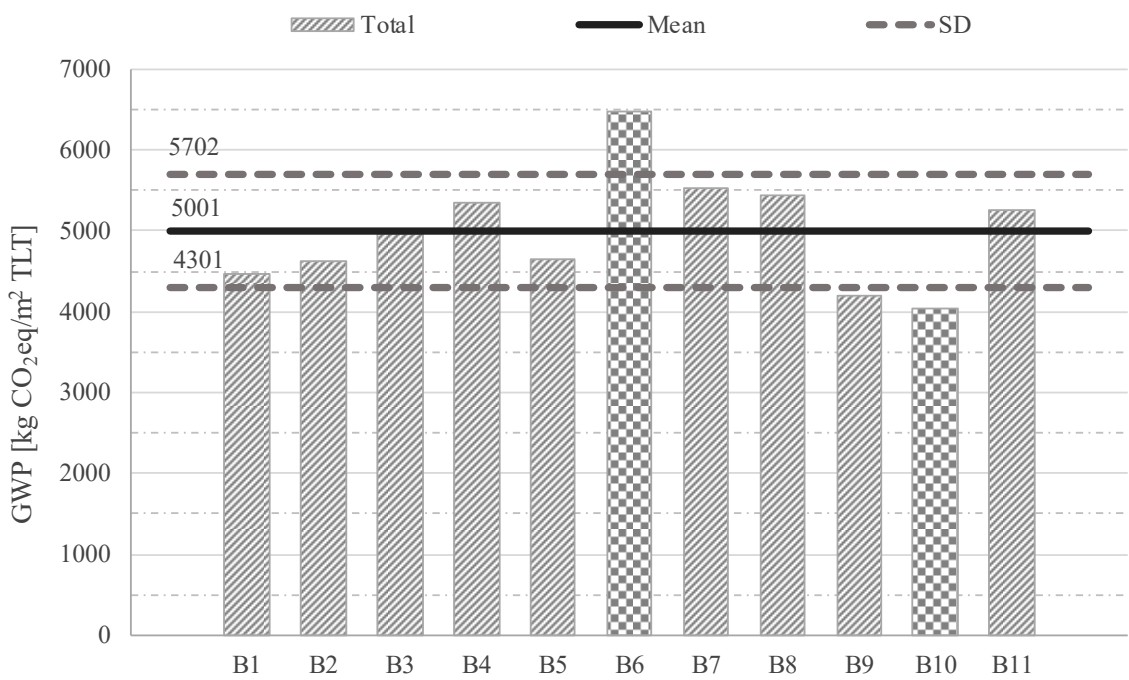

**Figure 6.** Normalised GWP index for all 11 buildings in total lifetime where TLT = 60 years.

The statistical variance of the normalised GWP index is also shown in Figure 7. The median of the analysed dataset was equal to 4965 kgCO$_2$eq/m$^2$TLT and, therefore, is close to the average value. Moreover, no values beyond the minimum and maximum values represented by whiskers were found, which indicates that the GWP index calculated for all buildings is not significantly diverse. Despite the differences in building characteristics, such as the usable area, number of floors, shape, type of windows, glazing ratio, HVAC systems, and energy source, as well as the different building components included in the LCI, the collected database of buildings showed a similar level of Global Warming Potential. As none of the values were defined as extreme data, results from all 11 buildings were used to develop the benchmark definition.

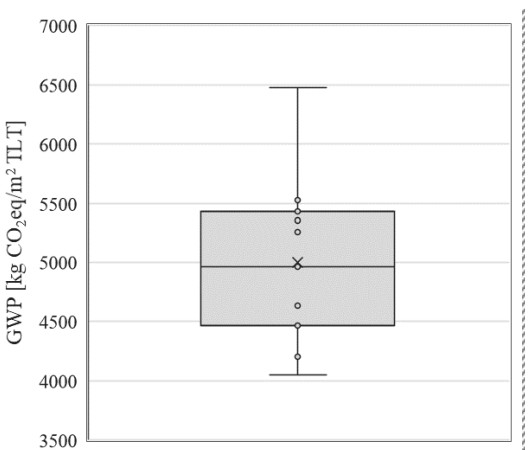

**Figure 7.** Statistical representation of the dataset.

Based on the results of the LCA study, a preliminary GWP benchmark for office buildings was defined in relation to the total Global Warming Potential normalised to the usable floor area. The calculated mean and median values for the 11 buildings were 5001 and 4965 kgCO$_2$eq/m$^2$TLT, respectively. Based on those outcomes, a GWP value of 5000 kg CO$_2$eq/m$^2$ in TLT = 60 years was selected as a reference mean value for defining the GWP benchmark of office buildings in Poland.

Finally, building classes were calculated according to the methodology defined by the European standard EN ISO 52003-1 [38], which sets the requirements and is used as a rating method to assess the energy performance of buildings. The defined building rating labels can be visualised in Table 9.

**Table 9.** Building rating labels.

| Building Class | GWP [kg $CO_2$eq/m$^2$TLT] | |
|:---:|:---:|:---:|
| A | ≤ | 1800 |
| B | ≤ | 2500 |
| C | ≤ | 3500 |
| D | ≤ | 5000 |
| E | ≤ | 7100 |
| F | ≤ | 10,000 |
| G | ≤ | 14,100 |

The collected database, consisting of data from 11 office buildings located in Poland, defines the average building class labelled as class D. The created database reflects a variety of typologies and technologies present in real buildings and thus reflects actual Polish building standards in terms of energy efficiency. In order to achieve a higher rating, future buildings will need to face more demanding requirements. As the energy demand has been already minimised through a high level of thermal protection and energy-efficient systems, a further decrease in energy use might be difficult to achieve or might not be economically justified. A more effective method for enhancing building standards will be therefore related to the fuel types and quantities used. A reduction in the GWP index could be done by on-site production of green energy by using the less emissive type of fuels or energy sources and by increasing the share of RES (renewable energy sources) in the Polish energy mix, which would consequently decrease the overall energy balance. Further improvement will involve the use of more ecological and environmentally friendly products, characterised by a longer service lifetime and a high level of recyclability.

## 5. Discussion and Future Work

In the recast EPBD, a commitment to developing a sustainable energy system by 2050 was stated [4]. This goal cannot be achieved without the decarbonisation of building stock, which requires in-build and operational greenhouse gas emissions to be considered. This study showed that three elements in building LCA are significant: building energy performance, embodied GHG, and energy mix. We showed that the energy used during the operation phase of a building's lifetime accounts for the highest proportion of the total emissions. The second highest value is related to materials, for which the quantity of embodied GHG emissions is influenced by the use of natural and environmentally friendly products. The last element, which affects both of those mentioned earlier, is the energy mix. The ratio of renewable and cleaner energy sources will most probably increase, changing the energy mix and, as a consequence, the embodied carbon will have a greater influence on the environmental impact of buildings. Therefore, the subsequent study should involve a sensitivity analysis that considers the energy mix, building technical and energy standard, HVAC system efficiency, and the use of natural and ecological materials.

In this paper, the LCI of the selected buildings differed, and in future research, the influence of the building elements scope included in the LCA on the GWP indicator should be verified and clearly defined. Also, as presented in [24], it is assumed that the total building lifetime has an influence on the final result. Thus, the national benchmark determination methodology must be preceded with an analysis of TLT, leading to the most appropriate time period being defined.

That final result was influenced by the number of building life stages included in the calculation. The first system that introduced a rating procedure according to assessed building life stages on the total carbon footprint was presented in Norway. The guidelines and definitions were based on the life

cycle analysis [34,35]. The zero-emission building (ZEB) definition has been specified for 6 ambition levels: ZEB-O÷EQ, ZEB-O, ZEB-OM, ZEB-COM, ZEB-COME, and ZEB-COMPLETE. The lowest level (ZEB-O÷EQ) considers energy use for operation (O), except energy use for equipment and appliances (EQ). The highest rating takes into account the total life cycle emissions. The Norway method is focused on achieving zero emissions by using on-site renewable energy generation to compensate for building emissions. Further work with a larger LCA database could be done to define similar levels for Polish office building stock.

## 6. Conclusions

The main goal of this study was to present the carbon footprint of newly built office buildings located in Poland with a preliminary definition of the possible building benchmark based on the GWP index. Although the created building database is not statistically representative, it shows the current state of the level of global warming potential of buildings designed and constructed according to current legislation, energy requirements, and technology. The outcomes of this study provide data regarding a benchmark definition that is consistent with the idea of the circular economy concept and decarbonisation requirements of the EPBD.

The results show that even a limited database of LCA results can be used to estimate a preliminary benchmark for new buildings. However, the limitation did not allow us to recognize the presented results as the final benchmark. The presented methodology can be used in future work where the LCA database is supplemented with results for additional buildings. Thus, there is a need to collect LCA results, which can be done using environmental certification schemes like BREEAM. Currently, most of the new commercial buildings in Poland are certified, and designers must enforce LCA among other analyses performed as part of environmental certification.

To ensure that future results are based on better data quality, some assumptions must be met. The building TLT must be agreed on and at the same level, as in different LCAs, this value can vary a lot, and as presented in [25], the adopted TLT has a noticeable influence on the final result. Also, the LCI must be standardized to include the same building elements in the analysis.

Even if the national policy does not include the GWP benchmark for new buildings, the estimated index values and proposed classification can be used by designers in early planning. The study presented in this paper provides a reference that can be used to achieve the ambitious goal of building stock decarbonisation in the next 30 years.

**Author Contributions:** J.R., A.K. and J.K.—concept; J.K.—introduction; J.R. and A.K.—methodology; J.R., A.K. and J.K.—calculations and benchmark proposition. All authors have read and agreed to the published version of the manuscript.

**Funding:** This research was co-financed by the Strategic Research Project of the Warsaw University of Technology "Circular Economy".

**Conflicts of Interest:** The authors declare no conflict of interest.

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
