# Peer review of "Preliminary Study on the GWP Benchmark of Office Buildings in Poland Using the LCA Approach"

_energies, doi:10.3390/en13133298_

Round 1

Reviewer 1 Report

The paper presents a benchmark of procedure and a rating system for the classification of office buildings in Poland based on GWP. The topic of the article is interesting but in my opinion there some issues that need to be addressed. 

First of all the authors refer to LCA but the analysis is based only on GWP, one impact indicator. Some references, already cited by the authors,  analyse LCA of buildings including a larger set of indicators. I suggest to fill this gap or to justify the current choice or to refer more to GWP than LCA (for example in the title).

English language needs to be improved. I suggest a deep revision  of the manuscript checking typos.

In figure 1 please insert the y-axis title

Author Response

Thank you for giving us the opportunity to submit a revised manuscript. We are grateful for your insightful comments to our paper. We have been able to incorporate changes to reflect most of your suggestions. We have highlighted all of the changes suggested by all of the reviewers within the manuscript. Here is a point-by-point response to your remarks and comments: 

The paper presents a benchmark of procedure and a rating system for the classification of office buildings in Poland based on GWP. The topic of the article is interesting but in my opinion there some issues that need to be addressed. First of all the authors refer to LCA but the analysis is based only on GWP, one impact indicator. Some references, already cited by the authors,  analyse LCA of buildings including a larger set of indicators. I suggest to fill this gap or to justify the current choice or to refer more to GWP than LCA (for example in the title). 

  • The title has been changed – the benchmark procedure is only based GWP index, because it is the most common and recognizable environmental indicator. 

In figure 1 please insert the y-axis title 

  • A figure 1 has been changed. The previous figure was too general and current is based on literature data. 

English language needs to be improved. I suggest a deep revision  of the manuscript checking typos. 

  • The article has been checked and corrected by MDPI – language proofreading. 

Reviewer 2 Report

This article aims to review the preliminary GHG emission benchmark for office buildings in Poland. The authors reviewed that the energy demand in the operational phase for the analysis of GHG emission in office buildings. For the detailed analysis of designbuilder simulations, the input data were clearly addressed in this manuscript. Also the partial efficiency of HVAC system should be addressed based on the replacement schedule of equipment during TLT 60 years. The authors need to describe the effects of heating and cooling sources on the GWP index more clearly.  

Author Response

Thank you for giving us the opportunity to submit a revised manuscript. We are grateful for your insightful comments to our paper. We have been able to incorporate changes to reflect most of your suggestions. We have highlighted all of the changes suggested by all of the reviewers within the manuscript. Here is a point-by-point response to your remarks and comments: 

This article aims to review the preliminary GHG emission benchmark for office buildings in Poland. The authors reviewed that the energy demand in the operational phase for the analysis of GHG emission in office buildings. For the detailed analysis of DesignBuilder simulations, the input data were clearly addressed in this manuscript. Also, the partial efficiency of HVAC system should be addressed based on the replacement schedule of equipment during TLT 60 years.  

  • Ithline 473 was added HVAC system efficiencyThe analysis was based on current emission factors (for materials and energy) and currents HVAC efficiencies, which were assumed to be constant. This is a common approach in building life cycle assessment 

The authors need to describe the effects of heating and cooling sources on the GWP index more clearly.   

  • The GWP index for energy sources was added. The benchmark definition was based on real case buildings and heating and cooling sources were defined according to technical documentation. The change of energy sources on GWP index was not the scope of this analysis. Effects of heating and cooling sources could be the part of future work. 
  • The article has been checked and corrected by MDPI – language proofreading. 

Reviewer 3 Report

The paper presents a study regarding the carbon footprint of newly built office buildings located in Poland with a preliminary definition of the possible building benchmark based on the Global Warming Potential index.

The tackled topic in this paper is worth to be examined, the sections are clear and connected. In terms of writing, I think the manuscript generally well-written. Overall, the introduction clearly states the background, problematic, and the main objective and the conclusion did summarize the findings of this work well.

However, I don’t think the paper is ready for publication unless some amendments are performed, namely the first two major comments, as follow:

  • Section 2.3: The excluding of the lowest and highest buildings is not accurate. To test the statistical significance of the extreme data, the authors should perform an outlier detection (please for more details check the article below). In case they detect outliers, the authors can exclude them from the sample, otherwise just excluding the highest and fewest is a false assumption. So, the authors should do this analyses and the remaining analyses should be correlated to the results of this investigation. For instance, as we can see in Figure 7 using the box plot, in the case of 11 buildings, the lowest and highest GWP are not outliers, so they must not be ignored when setting the benchmark.
    • Combined use of dynamic building simulation and metamodeling to optimize glass façade design for thermal comfort, Building and Environment 157 (2019), 47–63
    • Evaluation of the causes and impact of outliers on residential building energy use prediction using inverse modeling, Building and Environment 138 (2018), 194–206
  • There are several types of Benchmarking techniques, the authors should note this and talk briefly about them in section 2.3, then describe which type they used in the paper and why.
  • Figure 1: Do the authors produce the figure based on collected data from the literature? If it is the case, then some information about the data should be provided in the manuscript. If not, then a citation after the figure caption is required to show the readers the origin of the presented data.
  • Figure 1: what is the unit in the y-axis?
  • Figure 2: the numbering is false, please fix the number.
  • Line 355: I propose to add a figure illustrating the correlation between the mass of building material and the GWP.
  • Minor English issues can be found and a proofreading is required.

Author Response

Thank you for giving us the opportunity to submit a revised manuscript. We are grateful for your insightful comments to our paper. We have been able to incorporate changes to reflect most of your suggestions. We have highlighted all of the changes suggested by all of the reviewers within the manuscript. Here is a point-by-point response to your remarks and comments: 

The paper presents a study regarding the carbon footprint of newly built office buildings located in Poland with a preliminary definition of the possible building benchmark based on the Global Warming Potential index. 

The tackled topic in this paper is worth to be examined, the sections are clear and connected. In terms of writing, I think the manuscript generally well-written. Overall, the introduction clearly states the background, problematic, and the main objective and the conclusion did summarize the findings of this work well. 

However, I don’t think the paper is ready for publication unless some amendments are performed, namely the first two major comments, as follow: 

Section 2.3: The excluding of the lowest and highest buildings is not accurate. To test the statistical significance of the extreme data, the authors should perform an outlier detection (please for more details check the article below). In case they detect outliers, the authors can exclude them from the sample, otherwise just excluding the highest and fewest is a false assumption. So, the authors should do this analyses and the remaining analyses should be correlated to the results of this investigation. For instance, as we can see in Figure 7 using the box plot, in the case of 11 buildings, the lowest and highest GWP are not outliers, so they must not be ignored when setting the benchmark.  

Combined use of dynamic building simulation and metamodeling to optimize glass façade design for thermal comfort, Building and Environment 157 (2019), 47–63 

Evaluation of the causes and impact of outliers on residential building energy use prediction using inverse modeling, Building and Environment 138 (2018), 194–206 

  • The suggested corrections have been applied in section 2.3 and in section 4.2  

There are several types of Benchmarking techniques, the authors should note this and talk briefly about them in section 2.3, then describe which type they used in the paper and why.

  • A short description of different benchmarking approaches has been added in section 2.3, with the explanation which approach and why was chosen in further analysis. 

Figure 1: Do the authors produce the figure based on collected data from the literature? If it is the case, then some information about the data should be provided in the manuscript. If not, then a citation after the figure caption is required to show the readers the origin of the presented data. 

Figure 1: what is the unit in the y-axis? 

  • A figure 1 has been changed and a reference to the origin of the presented data has been given. 

 Figure 2: the numbering is false, please fix the number. 

  • A figure numbering was corrected. 

Line 355: I propose to add a figure illustrating the correlation between the mass of building material and the GWP. 

  • A figure was added. 

 Minor English issues can be found and a proofreading is required. 

  •  The article has been checked and corrected by MDPI – language proofreading. 

Round 2

Reviewer 2 Report

The authors changed the manuscript based on the comments. 

Reviewer 3 Report

I would like to thank the authors for their effort to improve the manuscript. The authors added some analysis and clarifications to consider my comments alongside English editing, which significantly improved the manuscript.